# Fully Cross-Attention Transformer for Guided Depth Super-Resolution

**DOI:** 10.3390/s23052723

**Published:** 2023-03-02

**Authors:** Ido Ariav, Israel Cohen

**Affiliations:** Andrew and Erna Viterbi Faculty of Electrical and Computer Engineering, Technion—Israel Institute of Technology, Haifa 3200003, Israel

**Keywords:** super-resolution, deep learning, depth maps, attention, multimodal, transformers

## Abstract

Modern depth sensors are often characterized by low spatial resolution, which hinders their use in real-world applications. However, the depth map in many scenarios is accompanied by a corresponding high-resolution color image. In light of this, learning-based methods have been extensively used for guided super-resolution of depth maps. A guided super-resolution scheme uses a corresponding high-resolution color image to infer high-resolution depth maps from low-resolution ones. Unfortunately, these methods still have texture copying problems due to improper guidance from color images. Specifically, in most existing methods, guidance from the color image is achieved by a naive concatenation of color and depth features. In this paper, we propose a fully transformer-based network for depth map super-resolution. A cascaded transformer module extracts deep features from a low-resolution depth. It incorporates a novel cross-attention mechanism to seamlessly and continuously guide the color image into the depth upsampling process. Using a window partitioning scheme, linear complexity in image resolution can be achieved, so it can be applied to high-resolution images. The proposed method of guided depth super-resolution outperforms other state-of-the-art methods through extensive experiments.

## 1. Introduction

High-resolution (HR) depth information of a scene plays a significant part in many applications, such as 3D reconstruction [1], driving assistance [2], and mobile robots. Nowadays, depth sensors such as LIDAR or time-of-flight cameras are becoming more widely used. However, they often suffer from low spatial resolution, which does not always suffice for real-world applications. Thus, ongoing research has been done on reconstructing a high-resolution depth map from a corresponding low-resolution (LR) counterpart in a process termed depth super-resolution (DSR).

The LR depth map does not contain the fine details of the HR depth map, so reconstructing the HR depth map can be challenging. Bicubic interpolation, for example, often produces blurry depth maps when upsampling the LR depth, which limits the ability to, e.g., separate between different objects in the scene.

In recent years, many learning-based approaches based on elaborate convolutional neural network (CNN) architectures for DSR were proposed [3,4,5,6,7]. These methods surpassed the more classic approaches such as filter-based methods [8,9], and energy minimization-based methods [10,11,12] in terms of computation speed and the quality of the reconstructed HR information. Although CNN-based methods improved the performance significantly compared with traditional methods, they still suffer from several drawbacks. To begin with, feature maps derived from a convolution layer have a limited receptive field, making long-range dependency modeling difficult. Second, a kernel in a convolution layer operates similarly on all parts of the input, making it content-independent and likely not the optimal choice. In contrast to CNN, transformers [13] have recently shown promising results in several vision-related tasks due to their use of attention. The attention mechanism enables the transformer to operate in a content-dependent manner, where each input part is treated differently according to the task.

LR depth information is often accompanied by HR color or intensity images in real-life situations. Thus, numerous methods proposed to use this HR image to guide the DSR process [3,4,7,14,15,16,17,18,19,20,21,22,23] since the HR image might provide some additional information that does not exist in the LR depth image, e.g., the edges of a color image can be used to identify discontinuities in a reconstructed HR depth image. However, one major limitation, termed texture-copying, still exists in these guided DSR methods. Texture copying may occur when intensity edges do not correspond to depth discontinuities in-depth maps, for example, a flat and highly textured surface. Consequently, the reconstruction of HR depth is then degraded due to the over-transfer of texture information.

This paper proposes a novel, fully transformer-based architecture for guided DSR. Specifically, the proposed architecture consists of three modules: shallow feature extraction, deep feature extraction and fusion, and an upsampling module. In this paper, we term the feature extraction and fusion module the cross-attention guidance module (CAGM). The shallow feature extraction module uses convolutional layers to extract shallow features from LR depth and HR color images, which are directly fed to the CAGM to preserve low-frequency information. Next, several transformer blocks are stacked to form the CAGM, each operating in non-overlapping windows from the previous block. Guidance from the color image is introduced via a cross-attention mechanism. In this manner, guidance from the HR color image is seamlessly integrated into the deep feature extraction process. This enables the network to focus on salient and meaningful features and enhance the edge structures in the depth features while suppressing textures in the color features. Moreover, contrary to CNN-based methods, which can only use local information, transformer blocks can exploit the input image’s local and global information. This allows learning of structure and content from a wide receptive field, which is beneficial for SR tasks [24]. As a final step, shallow and deep features are fused in the upsampling module to reconstruct HR depth. Section 4 shows that the proposed architecture provides better visual results with sharper boundaries and better root mean square error (RMSE) values than competing guided DSR approaches. We also show how the proposed architecture helps to mitigate the texture-copying problem of guided DSR. The proposed architecture is shown in Figure 1.

Our main contributions are as follows: (1) We introduce a transformer-based architecture with a novel guidance mechanism that leverages cross-attention to seamlessly integrate guidance features from a color image to the DSR process. (2) Linear memory constraints make the proposed architecture applicable even for large inputs. (3) This architecture is not limited to a fixed input size, so it can be applied to a variety of real-world problems. (4) Our system achieves state-of-the-art results on several depth-upsampling benchmarks.

The remainder of this paper is organized as follows. A summary of related work is presented in Section 2. We describe our architecture for guided DSR in Section 3. Section 4 reports the results of extensive experiments conducted on several popular DSR datasets. Additionally, an ablation study is conducted. We conclude and discuss future research directions in Section 5.

## 2. Related Work

### 2.1. Guided Depth Map Super-Resolution

A number of methods for reconstructing the HR depth map only from LR depth have been proposed in earlier works for single depth map SR. ATGV-Net was proposed by [5] combining a deep CNN in tandem with a variational method designed to facilitate the recovery of the HR depth map. Reference [25] modeled the mapping between HR and LR depth maps by utilizing densely connected layers coupled with a residual learning model. Auxiliary losses were tabulated at various scales to improve training.

However, it is pertinent to note that in most real-life scenarios, the LR depth image is coupled with a HR intensity image. Recently, several methods have been proposed to improve depth image quality, relying on the HR intensity image to guide the upsampling process. We group these methods under four sub-categories: filtering-based methods [26,27,28], global optimization-based methods [10,11,12,16,29,30,31,32,33,34], sparse representation-based methods [14,15], and deep learning-based methods [3,4,7,17,18,19,20,21,22,23,35,36,37,38,39,40], which are the focus of this paper.

Notable amongst the more classical works are [10], which formulated the upsampling of depth as a convex optimization problem. The upsampling process was guided by a HR intensity image. A bimodal co-sparse analysis was presented in [14] to describe the interdependency of the registered intensity and depth information. Reference [15] proposed a multi-scale dictionary as a method for depth map refinement, where local patches were represented in both depth and color via a combination of select basis functions.

Deep learning methods for SR of depth images have gained increasing attention due to recent success in SR of color images. A fully convolutional network was proposed in [35] to estimate the HR depth. To optimize the final result, this HR estimation was fed into a non-local variational model. Reference [4] proposed an “MSG-Net”, in which both LR (depth) and HR (color) features are combined within the high-frequency domain using a multi-scale fusion strategy. Reference [3] proposed extracting hierarchical features from depth and color images by building a multi-scale input pyramid. The hierarchical features are further concatenated to facilitate feature fusion, whilst the residual map between the reconstructed and ground truth HR depth is learned with a U-Net architecture. Reference [37] proposed another multi-scale network in which the LR depth map upsampling, guided by the HR color image, was performed in stages. Global and local residual learning is applied within each scale. Reference [17] proposed a cosine transform network in which features from both depth and color images were extracted using a semi-coupled feature extraction module. To improve depth upsampling, edges were highlighted by an edge attention mechanism operating on color features. Reference [19] proposed to use deformable convolutions [41] for the upsampling of depth maps, using the features of the HR guidance image to determine the spatial offsets. Reference [42] also applied deformable convolutions to enhance depth features by learning the corresponding feature of the high-resolution color image. An adaptive feature fusion module was used to fuse different level features adaptively. A network based on residual channel attention blocks was proposed in [20], where feature fusion blocks based on spatial attention were utilized to suppress texture-copying. Reference [21] proposed a progressive multi-branch aggregation design that gradually restores the degraded depth map. Reference [22] proposed separate branches for HR color image and LR depth map. A dual-skip connection structure, together with a multi-scale fusion strategy, allowed for more effective features to be learned. Reference [39] used a transformer module to learn the useful content and structure information from the depth maps and the corresponding color images, respectively. Then, a multi-scale fusion strategy was used to improve the efficiency of color-depth fusion. Reference [43] proposed explicitly incorporating the depth gradient features in the DSR process. Reference [44] proposed PDR-Net, which incorporates an adaptive feature recombination module to adaptively recombine features from a HR color guidance image with features from the LR depth. Then, a multi-scale feature fusion module is used to fuse the recombined features using multi-scale feature distillation and joint attention mechanism. Finally, Reference [23] presented an upsampling method that incorporates the intensity image’s high-resolution structural information into a multi-scale residual deep network via a cascaded transformer module.

However, the methods above mostly fuse the guidance features with the depth features using mere concatenation. Moreover, most of these methods rely on CNN for feature extraction, which operates on a limited receptive field and lacks the expressive power of transformers. At the same time, we propose using a CAGM module, which leverages transformers to fuse and extract meaningful features from HR color and LR depth images, resulting in superior results, as shown in Section 4.

### 2.2. Vision Transformers

Transformers [13] have gained great success across multiple domains recently. Contributing to this success was their inherent attention mechanism, which enables them to learn the long-range dependencies in the data. This success led many researchers to adopt transformers to computer vision tasks, where they have recently demonstrated promising results, specifically in image classification [45,46,47], segmentation [47,48], and object detection [49,50].

To allow transformers to handle 2D images, an input image $I\in R^{H\times W\times C}$ is first divided into non-overlapping patches of size (P,P). Each patch is flattened and projected to a d-dimensional vector via a trainable linear projection, forming the patch embeddings $X\in R^{N\times d}$ where H,W are the height and width of the image, respectively, *C* is the number of channels, and $N=H\times W/P^{2}$ is the total number of patches. Finally, *N* is the effective input sequence length for the transformer encoder. Patch embeddings are enhanced with position embeddings to retain 2D image positional information.

In [13], a vanilla vision transformer encoder is constructed by stacking blocks of multi-head self-attention (MSA) and MLP layers. A residual connection is applied after every block, and layer normalization (LN) before every block. Given a sequence of embeddings $X\in R^{N\times d}$ with dimension d as input, a MSA block produces an output sequence $\bar{X}\in R^{N\times d}$ via
(1)$$\begin{array}{ll} Q & =XW_{Q}\;,K=XW_{K}\;,V=XW_{V}\\ A & =\mathrm{Softmax}(QK^{T}/\sqrt{d})\\ \bar{X} & =AV \end{array}$$
where $W_{Q}$, $W_{K}$, and $W_{V}$ are learnable matrices of size d×d that project the sequence X into keys, queries, and values, respectively. $\bar{X}$ is a linear combination of all the values in V weighted by the attention matrix A. In turn, A is calculated from similarities between the keys and query vectors.

Transformers derive their modeling capabilities from computing self-attention A and $\bar{X}$. Since self-attention has a quadratic cost in time and space, it cannot be applied directly to images as *N* quickly becomes unmanageable. As a result of this inherent limitation, modality-aware sequence length restrictions have been applied to preserve the model’s performance while restricting sequence length. Reference [45] showed that a transformer architecture could be directly applied to medium-sized image patches for different vision tasks. The aforementioned memory constraints are mitigated by this local self-attention.

Although the above self-attention module can effectively exploit intra-modality relationships in the input image, in a multi-modality setting, the inter-modality relationships, e.g., the relationships between different modalities, also need to be explored. Thus, a cross-attention mechanism was introduced in which attention masks from one modality highlight the extracted features in another. Contrary to self-similarity, wherein query, key, and value are based on similarities within the same feature array, in cross-attention, keys, and values are calculated from features extracted from one modality, while queries are calculated from the other. Formally, a MSA block using cross-attention is given by
(2)$$\begin{array}{cc} Q & =\hat{X}W_{Q}\;,K=XW_{K}\;,V=XW_{V} \end{array}$$
where X is the input sequence of one modality and $\hat{X}$ is the input sequence of the second modality. The calculation of attention matrix A and output sequence $\bar{X}$ remains the same.

## 3. Proposed Method

### 3.1. Formulation

A guided DSR method aims to establish the nonlinear relation between corresponding LR and HR depth maps. The process of establishing this nonlinear relation is guided by a HR color image. We denote the LR depth map as $D_{\mathrm{LR}}\in R^{H/s\times W/s}$ and the HR guidance color image as $I_{\mathrm{HR}}\in R^{H\times W}$, where *s* is a given scaling factor. The corresponding HR depth map $D_{\mathrm{HR}}\in R^{H\times W}$ can be found from:(3)$$\begin{array}{c} D_{\mathrm{HR}}=\hat{F}\left(D_{\mathrm{LR}},I_{\mathrm{HR}};\theta \right) \end{array}$$
where $\hat{F}$ represents mapping learned by the proposed architecture, and θ represents the parameters of the learned network. Although the scaling factor *s* is usually an exponent of 2, e.g., s=2,4,8,16, our upsampling module can perform upsampling for other scaling factors as well, making this architecture flexible enough for real applications.

### 3.2. Overall Network Architecture

Throughout the remainder of this paper, we denote the proposed architecture as the fully cross-attention transformer network (FCTN). As shown in Figure 1, the proposed architecture consists of three parts: a shallow feature extraction module, a deep feature extraction and guidance module called the cross-attention guidance module (CAGM), and an upsampling module. The CAGM extracts features from the LR depth image and guides the HR intensity image simultaneously.

Before we elaborate on the structure of each module, some significant challenges in leveraging transformers’ performance for visual tasks, specifically SR, need to be addressed. First, in real-life scenarios, images can vary considerably in scale. Transformer-based models, however, work only with tokens of a fixed size. Furthermore, to maintain HR information, SR methods avoid downscaling the input as much as possible. Processing HR inputs of this magnitude would be unfeasible for vanilla transformers due to computational complexity as described in Section 2.2.

#### 3.2.1. Shallow Feature Extraction Module

The proposed shallow feature extraction module extracts essential features to be fed to the CAGM. Shallow features are extracted from LR depth and HR color images via a single convolution layer with a kernel size of 3×3, followed by an activation function of a rectified linear unit (ReLU). In the experiments, we did not notice any noticeable improvement by using more than a single layer for shallow feature extraction. For shallow feature extraction, incorporating a convolution layer leads to more stable optimization and better results [51,52,53]. Moreover, the input space can also be mapped to a higher-dimensional feature space *d* easily.

Specifically, the shallow feature extraction module can be formulated as
(4)$$\begin{array}{cc} I_{0} & =\sigma ({\mathrm{Conv}}_{3}\left(I_{\mathrm{HR}}\right)) \end{array}$$
(5)$$\begin{array}{cc} D_{0} & =\sigma ({\mathrm{Conv}}_{3}\left(D_{\mathrm{LR}}\right)) \end{array}$$
where σ is a ReLU activation function and ${\mathrm{Conv}}_{3}\left(\cdot \right)$ is a 3×3 kernel.

#### 3.2.2. Deep Feature Extraction and Guidance Module

While shallow features primarily contain low frequencies, deep features recover lost high frequencies. We propose a stacked transformer module that extracts deep features from the LR depth image based on the work of [47]. Self(cross)-attention is computed locally within non-overlapping windows, with complexity linear with image size. Working with large and variable-sized inputs is made feasible due to the aforementioned linear complexity. In addition, we shift the windows partitioning into consecutive layers. Overlapping of the shifted and preceding layer windows causes neighboring patches to gradually merge, and thus modeling power is significantly enhanced. Overall, the transformer-based module can efficiently extract and encode distant dependencies needed for dense tasks such as SR.

In addition, motivated by [54], we employ global and local skip connections. By using long skip connections, low-frequency information can be transmitted directly to the upsampling module, helping the CAGM focus on high-frequency information and stabilize training [51]. Furthermore, it allows the aggregation of features from different blocks by using such identity-based connections.

Besides deep feature extraction, a practical guidance module is also required to enhance the deep features extracted from LR depth and exploit the inter-modality information from the available HR color image. Traditionally, CNN-based methods extract features from the color image and concatenate them with features extracted from the depth image in a separate branch to obtain guidance from the color image. All features handled via this guidance scheme are treated equally in both the spatial and channel domains. Furthermore, CNN-derived feature maps have a limited receptive field, affecting guidance quality. In comparison, we propose providing guidance from the HR color image by incorporating a cross-attention mechanism to the aforementioned feature extraction transformer module. Cross-attention is a novel and intuitive fusion method in which attention masks from one modality highlight the extracted features in another. In this manner, both the inter-modality and intra-modality relationships are learned and optimized in a unified model. Thus, in the proposed CAGM, the feature extraction process from the LR depth and guidance from the HR image are seamlessly integrated into a single branch. Guidance from the HR image is injected into every block in the feature extraction module, providing multi-stage guidance. In particular, guidance provided to the lower-level features passed through the long skip connections ensures that high-resolution information is preserved and passed to the upsampling module. Lastly, by incorporating the guidance in the form of cross-attention, long-range dependencies between the LR depth patches and the guidance image patches can be exploited for better feature extraction.

To exploit the HR information further, we feed the HR intensity image to a second cascaded transformer module termed color feature guidance (CFG) to extract even more valuable HR information. The CFG is based on self-attention only and aims to encode distant dependencies in the HR image. These features are then used to scale the features extracted from the CAGM by element-wise multiplication before feeding them to the upsampling module.

We note that contrary to common practice in vision tasks, no downsampling of the input is done throughout the network. This way, our architecture preserves as much high-resolution information as possible, albeit at a higher computational cost.

Formally, given $I_{0}$ and $D_{0}$, provided by the shallow feature extraction module as input, the CAGM applies *K* cross-attention transformer blocks (CATB). Every CATB is constructed from *L* cross-attention transformer layers (CATL), and a convolutional layer and residual skip connection are inserted at the end of every such block. Finally, a 3×3 convolutional layer is applied to the output of the last CATB. This last convolutional layer improves the later aggregation of shallow and deep features by bringing the inductive bias of the convolution operation into the transformer-based network. Furthermore, the translational equivariance of the network is enhanced. In addition, $I_{0}$ is fed to the CFG comprised of $\hat{L}$ transformer layers with self-attention. The CFG output is scaled to [0,1] using a sigmoid function and then used to scale the CAGM output before the upsampling module

The CFG module is formulated as
(6)$$\begin{array}{cc} {\hat{I}}_{i} & ={\mathrm{TL}}_{i}\left({\hat{I}}_{i-1}\right)\;,\;i=1\ldots \hat{L} \end{array}$$
(7)$$\begin{array}{cc} F_{\mathrm{CFG}} & ={\mathrm{Conv}}_{3}\left({\hat{I}}_{\hat{L}}\right)+I_{0} \end{array}$$
where ${\hat{I}}_{0}=I_{0}$ and TL stands for a vanilla transformer layer with self-attention. Finally, the entire CAGM can be formulated as
(8)$$\begin{array}{cc} \left(I_{i},D_{i}\right) & ={\mathrm{CATB}}_{i}\left(I_{i-1},D_{i-1}\right)\;,\;i=1\ldots K \end{array}$$
(9)$$\begin{array}{cc} F_{\mathrm{CAGM}} & ={\mathrm{Conv}}_{3}\left({\mathrm{CATB}}_{K}\right)⊗\hat{\sigma }\left(F_{\mathrm{CFG}}\right)+D_{0} \end{array}$$
where ⊗ is element-wise multiplication, ${\mathrm{Conv}}_{3}\left(\cdot \right)$ is a convolution layer with a 3×3 kernel and $\hat{\sigma }$ is a sigmoid function.

#### 3.2.3. Cross-Attention Transformer Layer

The proposed cross-attention transformer layer (CATL) is modified from the standard MSA block presented in [13]. The two significant differences are; First, we use a cross-attention mechanism instead of self-attention. We demonstrate the effectiveness of using a cross-attention mechanism in Section 4.4. Second, cross-attention is computed locally for each window, ensuring linear complexity with image size, which makes it feasible for large inputs to be handled, as is often the case in SR.

Given as input feature map $F\in R^{\hat{H}\times \hat{W}\times d}$ extracted from either color or depth images, we first construct $F_{\mathrm{win}}\in R^{\hat{H}\hat{W}/M^{2}\times M^{2}\times d}$ by partitioning F into M×M non-overlapping windows. Zero padding is applied during the partitioning process if necessary. Similarly to [55], relative position embeddings are added to $F_{\mathrm{win}}$ so that positional information can be retained. In a similar manner, the process is performed for both color and depth feature maps; we refer to this joint embedding as $Z_{I}^{0}$ and $Z_{D}^{0}$ for the color and depth, respectively.

In each CATL, the MSA module is replaced with a windows-based cross-attention MSA (MSA_ca_), while the other layers remain unchanged. By applying Equation (Equation 2) locally within each M×M window, we avoid global attention computations. Moreover, keys and values are calculated from the depth feature map, while the queries are calculated from the color feature map. Specifically, as illustrated in Figure 2b, our modified CATL consists of MSA_ca_ followed by a 2-layer MLP with GELU nonlinearity. Every MLP and MSA_ca_ module is preceded by an LN layer, and each module is followed by a residual connection.

To enable cross-window connections in consecutive layers, regular and shifted window partitionings are used alternately. In shifted window partitioning, features are shifted by M/2,M/2 pixels. Finally, the CATL can be formalized as
(10)$$\begin{array}{cc} \hat{Z} & ={\mathrm{MSA}}_{\mathrm{ca}}\left(\mathrm{LN}\left(Z_{I}^{0},Z_{D}^{0}\right)\right)+Z_{D}^{0} \end{array}$$
(11)$$\begin{array}{cc} Z & =\mathrm{MLP}\left(\mathrm{LN}\left({\hat{Z}}^{1}\right)\right)+{\hat{Z}}^{1} \end{array}$$
where $\hat{Z}$ and Z denote the output features of the MSA_ca_ and MLP modules, respectively.

#### 3.2.4. Upsampling Module

The upsampling module operates on the CAGM output, scaled via the CFG module, as elaborated in Section 3.2.2. It aims to recover high-frequency details and reconstruct the HR depth successfully. The CAGM output is first passed through a convolution layer followed by a ReLU activation function to aggregate shallow and deep features from the CAGM. Then, we use a pixel shuffle module [56] to upsample the feature map to the HR resolution. Each pixel shuffle module can perform upsampling by a factor of two or three, and we concatenate such modules according to the desired scaling factor. Finally, the upsampled feature maps are passed through another convolution layer that outputs the reconstructed depth. The parameters of the entire upsampling module are learned in the training process to improve model representation.

Formally, given the output of the CAGM module $F_{\mathrm{CAGM}}\in R^{H/s\times W/s}$, where *s* is the scaling factor, the upsampling module performs an upsampling by a factor *s* to reconstruct $D_{\mathrm{HR}}\in R^{H\times W}$. The upsampling process for a given *s* can be formulated as follows:(12)$$\begin{array}{ll} \; & F_{\mathrm{US},0}={\mathrm{Conv}}_{3}\left(F_{\mathrm{CAGM}}\right)\\ \; & F_{\mathrm{US},i}={\mathrm{PixellShuffle}}_{i}\left(F_{\mathrm{US},i-1}\right)\;,\;i=1\ldots \log_{2}\left(s\right)\\ \; & D_{\mathrm{HR}}={\mathrm{Conv}}_{3}\left(F_{\mathrm{US},i}\right)\;. \end{array}$$
where ${\mathrm{Conv}}_{3}\left(\cdot \right)$ is a convolution layer with a 3×3 kernel. More implementation details are given in Section 4.1.

## 4. Experiments

### 4.1. Training Details

We constructed train and test data similarly to [3,4,23,25] using 92 pairs of depth and color images from the MPI Sintel depth dataset [57] and the Middlebury depth dataset [58,59,60]. The training and validation pairs used in this study are similar to the ones used in [4,23]. We refer the reader to [57,58] for further information on the data included in the aforementioned datasets.

During training, we randomly extracted patches from the full-resolution images and used these as input to the network. We used an LR patch size of 96×96 pixels to reduce memory requirements and computation time since using larger patches had no significant impact on training accuracy. Consequently, we used HR patches of 192×192 and 384×384 for upsampling factors of 2 and 4, respectively. Given that some full-scale images had a full resolution of <400, we used LR patch sizes of 48×48 and 24×24 for upsampling factors 8 and 16, respectively. In order to generate the LR patches, each HR patch was downsampled with bicubic interpolation. As an augmentation method, we used a random horizontal flip while training.

### 4.2. Implementation Details

We construct the CAGM module in the proposed architecture by stacking K=6 CATBs. Each CATB is constructed from L=6 CATL modules as described in Section 3.2.2. These values for *K* and *L* provided the best performance to network size trade-off in the experiments, and Section 4.4, we report results with other configurations. All convolution layers have a stride of one with zero padding, so the features’ size remains fixed. Throughout the network, in convolution and transformer blocks, we use a feature (embedding) dimension size of d=64. We output depth values from the final convolution layer, which has only one filter. For window partitioning in the CATL, we use M=12, and each MSA module has six attention heads.

We used the PyTorch framework [61] to train a dedicated network for each upsampling factor s∈2,4,8,16. Each network was trained for $3\times 10^{5}$ iterations and optimized using the $L_{1}$ loss and the ADAM optimizer [62] with $\beta_{1}=0.9$, $\beta_{2}=0.999$ and $\epsilon =10^{-8}$. We used a learning rate of $10^{-4}$, dividing the learning rate by 2 for every $1\times 10^{5}$ iteration. All the models were trained on a PC with an i9-10940x CPU, 128GB RAM, and two Quadro RTX6000 GPUs.

### 4.3. Results

This section provides quantitative and qualitative evaluations of the proposed architecture for guided DSR. Our proposed architecture was evaluated on both the noise-free and the noisy Middlebury datasets. Further, we conduct experiments on the NYU Depth v2 dataset in order to demonstrate the generalization capabilities of the proposed architecture. We compare the results to other state-of-the-art methods, including global optimization-based methods [10,32], a sparse representation-based method [14], and mainly state-of-the-art deep learning-based methods [3,4,7,17,19,20,21,22,23,25,37,39,43,44]. We also report the results of naive bicubic interpolation as a baseline.

#### 4.3.1. Noise-Free Middlebury Dataset

The Middlebury dataset provides high-quality depth and color image pairs from several challenging scenes. First, we evaluate the different methods for the noise-free Middlebury RGB-D datasets for different scaling factors. In Table 1, we report the obtained RMSE values. Boldface indicates the best RMSE for each evaluation, while the underline indicates the second best. In Table 1, all results are calculated from upsampled depth maps provided by the authors or generated by their code.

Clearly, from Table 1 we conclude that deep learning-based methods [3,4,7,17,19,20,21,22,23,25,37] outperform the more classic methods for DSR. In terms of RMSE values, the proposed architecture provides the best performance across almost all scaling factors. For large scaling factors, e.g., 8,16, which are difficult for most methods, our method provides good reconstruction with the lowest RMSE error across all datasets. For scaling factors x4/x8/x16, our method obtained 0.48/0.99/1.55 as the average RMSE for the entire test set, respectively. Our results outperform the second-best results in terms of average RMSE values by 0.01/0.09/0.16, respectively.

In Figure 3 and Figure 4, we provide upsampled depth maps on the “Art” and “Moebius” datasets and a scale factor of 8 for qualitative evaluation. Upsampled depth maps are generated from 5 state-of-the-art methods, which are MSG [4], DSR [3], RDGE [32], RDN [7] and CTG [23]. We also provide bicubic interpolation as a baseline for comparison. Compared with competing methods, the proposed architecture provides more detailed HR depth boundaries. Additionally, our approach mitigates the texture-copying effect evident in some other methods, as shown by the red arrow. A significant factor contributing to these results is the attention mechanism built into the transformer model. This attention mechanism transfers HR information from the guidance image to the upsampling process in a sophisticated manner. Moreover, the transformer’s ability to consider both local and global information is key to improved performance at large scaling factors. Finally, these evaluations indicate that our CAGM contributes significantly to the success of depth map SR and enables accurate reconstruction even in complex scenarios with various degradations.

#### 4.3.2. Noisy Middlebury Dataset

We further demonstrate the robustness of the proposed architecture on the noisy Middlebury dataset. We added Gaussian noise to the LR training data, simulating the case where depth maps are corrupted during acquisition, in the same way as [3,7,23,37]. All the models were retrained and evaluated on a test set corrupted with the same Gaussian noise. For the noisy dataset, we report the RMSE values in Table 2.

Our first observation is that noise added to the LR depth maps significantly affects the reconstructed HR depth maps regardless of the method or scaling factor used. However, the proposed architecture still generates clean and sharp reconstructions and outperforms competing methods in terms of RMSE.

An even more realistic scenario is that data acquired by both the depth and color sensors are corrupted by noise. Our method was further tested by adding Gaussian noise with a mean of 0 and a standard deviation of 5 to the HR guidance images. This was done both in training and in testing. We again retrained the models and report the obtained average RMSE values in Table 3. In Table 3, we observe that the added noise in the HR guidance image did not significantly affect the performance of our method, compared to only adding noise to LR depth. According to our results, the proposed CAGM is somewhat insensitive to noise added to the guidance image.

#### 4.3.3. NYU Depth v2 Dataset

In this section, the proposed architecture is tested on the challenging public NYU Depth v2 [63] dataset as a means of demonstrating its generalization ability. There are 1449 high-quality RGB-D images of natural indoor scenes in this dataset, with apparent misalignments between depth maps and color images. We note that data from NYU Depth v2 are very different from the Middlebury Dataset and were not included in the training data of our models.

We report the average RMSE value across the entire dataset in Table 4. Boldface indicates the best RMSE value. As a baseline, we report the results of Bicubic interpolation as well as the results of competing guided SR approaches; ATGV-Net [5], MSG [4], DSR [3], RDN [7], RYN [20], PMBA [21], DEAF [42], JIIF [40], DCT [17], and CTG [23]. The proposed architecture achieves the lowest average RMSE, demonstrating the proposed method’s generalization ability and robustness.

#### 4.3.4. Inference Time

For a DSR method to applyto real-world applications, it is often required to work in a close-to-real-time performance. Thus, we report the inference time of the proposed architecture compared to other competing approaches. Inference times were measured using an image of size 1320×1080 pixels and the setup described in Section 4.2. We report our results in milliseconds in Table 5

Table 5 shows that compared to traditional methods, the proposed architecture, as well as other deep learning-based methods, provide significantly faster inference times. Moreover, the proposed method is comparable to competing methods and achieves lower RMSE values. In contrast, References [10,12,32] require multiple optimization iterations to obtain accurate reconstructions, leading to slower inference times. Some methods, such as [3,32], upsample the LR depth as an initial preprocessing step before the image is fed to the model. As a result, they show very similar inference times regardless of the scaling factor.

### 4.4. Ablation Study

In the ablation study, we test the effects of the CATB number in the CAGM and CATL number in each CATB on model performance. Results are shown in Figure 5a,b, respectively. It is observed that the RMSE of the reconstructed depth is positively correlated with both hyperparameters until it becomes eventually saturated. As we increase either hyperparameter, model size becomes increasingly prominent, and training\inference time and memory requirements are negatively impacted. Thus, to balance the performance and model size, we choose 6 for both hyperparameters as described in Section 4.2. CATL numbers were evaluated with a configuration of K=6 CATBs.

The impact of each component in our design is evaluated via the following experiments: (1) Our architecture without any guidance from the color image, denoted as “Depth-Only”. (2) Our architecture without shifted windows in the CATL, denoted “w/o shift”. (3) Our architecture without the CFG module, denoted “w/o CFG”. (4) Our architecture without the use of cross-attention for guidance. In this setting, we replaced the CATL with a similar design using only self-attention with depth features as input. Features from the color image were concatenated after every modified CATL to provide guidance. We denote this setting as “w/o cross-attention”.

We evaluate the different designs on the Middlebury test set at scaling factors 4, 8, and 16. We use the same CATB and CATL configuration as described in Section 4.2 in these experiments. We summarize the results in Table 6 and observe that: (1) As expected, using only the LR depth for DSR without guidance from a color image provides inferior results. (2) As also observed in [47], incorporating shifted window partitioning into our CATL improves the performance. Using shifted windows partitioning enables connections among windows in the preceding layers, improving the representation capability of each CATL. (3) Our CFG module provides additional high-frequency information directly to the upsampling module. As a result, the upsampling module can reconstruct a higher quality HR depth, and we observe that performance improves slightly. (4) We observe that using a simple concatenation of features instead of the proposed cross-attention guidance leads to inferior results. Incorporating the guidance from the color image via cross-attention allows the color feature to interact elaborately with the depth features and to encode long-distant dependencies between the two modalities.

## 5. Conclusions

We introduce a novel transformer-based architecture with cross-attention for guided DSR. First, a shallow feature extraction module extracts meaningful features from LR depth and HR color images. These features are fed to a cascaded transformer module with cross-attention, which extracts more elaborate features while simultaneously incorporates guidance from the color features via the cross-attention mechanism. The cascaded transformer module is constructed by stacking transformer layers with shifted window partitioning, which enables interactions between windows in consecutive layers. Using such a design, the proposed architecture achieves state-of-the-art results on the DSR benchmarks. At the same time, model size and inference time remain comparably small, making our architecture usable for real-world applications.

Our future work will explore more realistic depth artifacts (e.g., sparse depth values, misalignment between guidance and depth images, etc.). Moreover, we will examine the proposed architecture on additional real-world continuous data acquired from sensors mounted, e.g., on an autonomous robot.

## Figures and Tables

**Figure 1 sensors-23-02723-f001:**
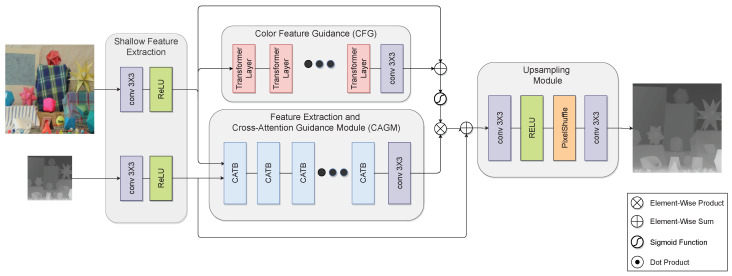
The proposed FCTN architecture for guided depth SR with a 2× upsampling factor.

**Figure 2 sensors-23-02723-f002:**
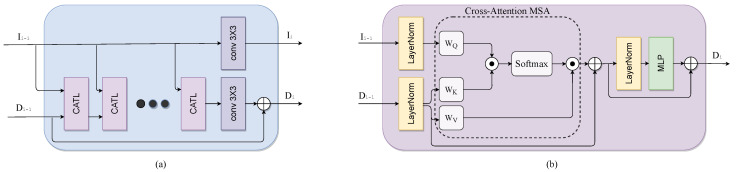
(**a**) Cross-Attention Transformer Block. (**b**) Cross-Attention Transformer Layer.

**Figure 3 sensors-23-02723-f003:**
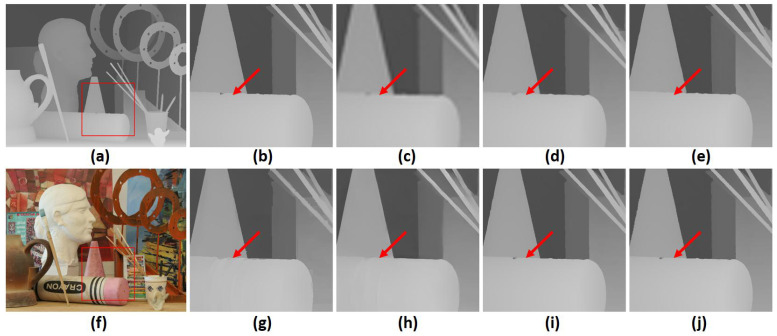
A visual quality comparison for depth map SR at a scale factor of 8 on the noise-free “art” dataset. (**a**) HR depth image, (**f**) HR color image, (**b**) extracted ground truth patch (marked by a red square), and upsampled patches by (**c**) Bicubic, (**d**) MSG [4], (**e**) DSR [3], (**g**) RDGE [32], (**h**) RDN [7], (**i**) CTG [23], (**j**) the proposed FCTN method (best viewed on the enlarged electronic version).

**Figure 4 sensors-23-02723-f004:**
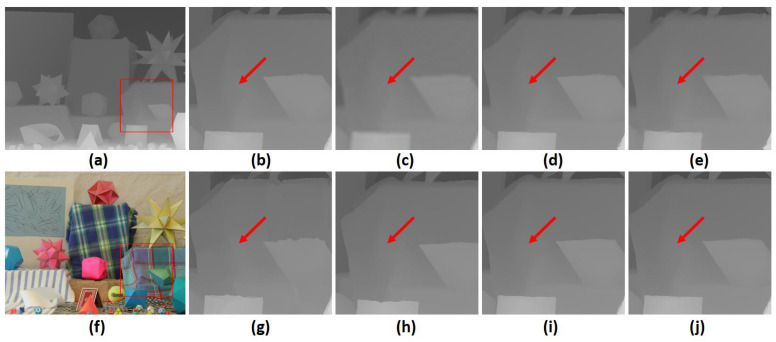
A visual quality comparison for depth map SR at a scale factor of 8 on the noise-free “Moebius” dataset. (**a**) HR depth image, (**f**) HR color image, (**b**) extracted ground truth patch (marked by a red square, and upsampled patches by (**c**) Bicubic, (**d**) MSG [4], (**e**) DSR [3], (**g**) RDGE [32], (**h**) RDN [7], (**i**) CTG [23], (**j**) the proposed FCTN method (best viewed on the enlarged electronic version).

**Figure 5 sensors-23-02723-f005:**
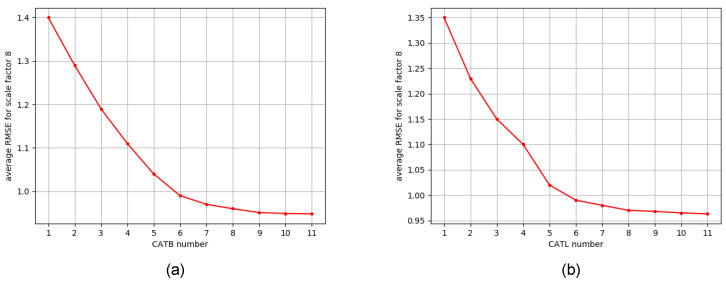
Ablation study on different configurations of the proposed CAGM. Results are the average RMSE on the noise-free Middlebury dataset for scaling factor 8. (**a**) The effect of the CATB number in the CAGM, and (**b**) the effect of CATL number in each CATB.

**Table 1 sensors-23-02723-t001:** An analysis of RMSE Values for different scaling factors on the noise-free Middlebury dataset. Boldface indicates the best RMSE for each evaluation, while the underline indicates the second best.

Method	Art			Books			Laundry			Dolls			Moebius			Reindeer		
	x4	x8	x16	x4	x8	x16	x4	x8	x16	x4	x8	x16	x4	x8	x16	x4	x8	x16
Bicubic	3.88	5.60	8.58	1.56	2.24	3.36	2.11	3.10	4.47	1.21	1.78	2.57	1.40	2.05	2.95	2.51	3.92	5.72
TGV [10]	4.06	5.08	7.61	2.21	2.47	3.54	2.20	3.92	6.75	1.42	2.05	4.44	2.03	2.58	3.50	2.67	4.29	8.80
JID [14]	1.92	2.76	5.74	0.71	1.01	1.93	1.10	1.83	3.62	0.92	1.26	1.74	0.89	1.27	2.13	1.41	2.12	4.64
RDGE [32]	3.26	4.31	6.78	1.53	2.18	2.92	2.06	2.87	4.22	1.49	1.94	2.45	1.44	2.21	2.79	2.58	3.24	4.90
MSG [4]	1.49	2.79	5.95	0.66	1.09	1.87	1.02	1.35	2.03	0.72	0.99	1.59	0.68	1.14	2.07	1.33	1.72	2.99
DSR [3]	1.21	2.23	3.95	0.60	0.89	1.51	0.75	1.21	1.89	0.81	1.10	1.60	0.67	0.96	1.57	0.96	1.57	2.54
PSR [25]	1.59	2.57	4.83	0.83	1.19	1.70	0.92	1.52	2.97	0.91	1.31	1.88	0.86	1.21	1.87	1.11	1.80	3.11
MFR [37]	1.54	2.71	4.35	0.63	1.05	1.78	1.11	1.75	3.01	0.89	1.22	1.74	0.72	1.10	1.73	1.23	2.06	3.74
RDN [7]	1.47	2.60	4.16	0.62	1.00	1.68	0.96	1.63	2.86	0.88	1.21	1.71	0.69	1.06	1.65	1.17	1.60	3.58
PMBA [21]	1.19	2.47	4.37	0.53	1.10	1.51	0.80	1.54	2.72	0.66	1.08	1.75	0.55	1.13	1.62	0.92	1.76	2.86
RYN [20]	0.98	2.04	3.37	0.36	0.73	1.37	0.64	1.21	2.01	0.59	0.97	**1.37**	0.50	0.81	1.37	0.74	1.41	2.22
CUN [22]	1.05	2.27	3.67	0.35	0.73	1.45	0.59	1.15	2.25	0.61	0.97	1.43	0.48	0.77	1.31	0.82	1.51	2.38
GDC [19]	1.09	2.04	3.58	0.38	0.68	1.41	0.64	1.13	2.13	0.63	0.97	1.44	0.49	0.79	1.37	0.84	1.51	2.43
TDTN [39]	1.24	2.45	–	0.48	0.86	–	0.68	1.29	–	0.76	1.15	–	0.61	0.91	–	0.95	1.75	–
MIG [43]	1.46	2.74	4.26	0.58	0.95	1.67	0.93	1.57	2.85	0.87	1.21	1.75	0.66	1.04	1.66	1.17	2.11	3.81
PDR [44]	1.59	2.57	4.83	0.83	1.19	1.70	0.92	1.52	2.97	0.91	1.31	1.88	0.86	1.21	1.87	1.11	1.80	3.11
CTG [23]	0.73	1.89	2.76	0.35	**0.66**	1.22	**0.43**	0.87	1.62	0.50	0.90	1.49	**0.46**	0.76	1.31	**0.43**	1.19	1.84
FCTN (Proposed)	**0.71**	**1.71**	**2.56**	**0.34**	0.68	**1.12**	0.47	**0.79**	**1.43**	**0.45**	**0.81**	1.40	**0.46**	**0.68**	**1.18**	0.47	**1.12**	**1.64**

**Table 2 sensors-23-02723-t002:** An analysis of RMSE values for different scaling factors on the noisy Middlebury dataset. Boldface indicates the best RMSE for each evaluation, while the underline indicates the second best.

Method	Art		Books		Laundry		Dolls		Moebius		Reindeer	
	x8	x16	x8	x16	x8	x16	x8	x16	x8	x16	x8	x16
Bicubic	6.74	9.04	4.68	5.30	5.35	6.53	4.51	4.90	4.54	5.02	5.71	7.12
TGV [10]	7.26	12.05	2.88	4.73	4.45	8.06	2.82	5.14	3.01	6.11	4.65	9.03
MSG [4]	4.24	7.42	2.48	4.19	3.31	4.88	2.53	3.41	2.47	3.76	3.36	4.95
MFR [37]	3.97	6.14	2.13	3.17	2.82	4.57	2.25	3.30	2.13	3.33	3.01	4.86
RDN [7]	4.09	6.62	2.11	3.36	2.88	5.11	2.33	3.59	2.18	3.69	3.09	4.93
DSR [3]	–	6.96	–	5.66	–	7.54	–	4.28	–	3.39	–	5.25
RYN [20]	3.47	–	1.88	–	2.47	–	1.97	–	1.87	–	2.68	–
GDC [19]	3.31	4.77	1.69	**2.46**	2.20	3.36	1.89	2.59	1.72	2.68	2.57	3.44
JIIF [40]	3.87	7.14	1.75	2.47	–	–	–	–	2.03	3.18	–	–
MIG [43]	3.95	6.15	2.10	3.17	3.00	4.88	2.21	3.51	2.12	3.51	3.04	4.97
CTG [23]	3.26	4.72	1.61	2.96	1.63	3.47	1.64	**2.16**	1.63	2.24	**1.79**	3.59
FCTN (Proposed)	**3.01**	**4.55**	**1.54**	2.66	**1.61**	3.15	**1.59**	2.32	**1.27**	**2.09**	1.81	**3.17**

**Table 3 sensors-23-02723-t003:** An Analysis of the average RMSE values for different noise schemes.

Middlebury Dataset Version	x4	x8	x16
Noise-Free	0.48	0.99	1.55
Depth Noise	1.17	1.80	2.99
Depth and Color Noise	1.35	2.01	3.19

**Table 4 sensors-23-02723-t004:** Quantitative comparisons of the ablation experiments. Reported results are average RMSE on the noise-free Middlebury dataset for scaling factors 4, 8, and 16. Boldface indicates the best RMSE for each evaluation, while the underline indicates the second best.

Method	Average RMSE on NYU Depth v2 Dataset
Bicubic	2.36
ATGV-Net [5]	1.28
MSG [4]	1.31
RDN [7]	1.21
DSR [3]	1.34
RYN [20]	1.06
PMBA [21]	1.06
DEAF [42]	1.12
JIIF [40]	1.37
DCT [17]	1.59
CTG [23]	0.95
FCTN (Proposed)	**0.91**

**Table 5 sensors-23-02723-t005:** Average inference times (milliseconds) for different scaling factors.

Method	x2	x4	x8	x16
Bicubic	10	10	10	10
TGV [10]	45,730	49,780	46,340	46,200
AR [12]	158,010	157,730	157,950	158,770
RDGE [32]	68,070	67,690	68,450	68,170
MSG [4]	260	300	380	420
DSR [3]	220	230	230	230
RYN [20]	460	630	720	880
CTG [23]	150	380	480	530
FCTN (Proposed) [23]	140	304	420	490

**Table 6 sensors-23-02723-t006:** An analysis of the average RMSE values for different ablation experiments on the noise-free Middlebury dataset. Boldface indicates the best RMSE for each evaluation.

Design	Depth-Only			w/o Shift			w/o CFG			w/o Cross-Attention			FCTN (Proposed)		
Scale Factor	x4	x8	x16	x4	x8	x16	x4	x8	x16	x4	x8	x16	x4	x8	x16
RMSE	0.65	1.39	3.01	0.52	1.14	1.90	0.51	1.06	1.79	0.59	1.28	2.17	**0.48**	**0.99**	**1.55**

## Data Availability

Publicly available datasets were analyzed in this study. This data can be found here: MPI Sintel Dataset—[http://sintel.is.tue.mpg.de/, accessed on 15 February 2023]. Middlebury Stereo Datasets—[https://vision.middlebury.edu/stereo/data/, accessed on 15 February 2023].

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
