# Peer review of "Fully Cross-Attention Transformer for Guided Depth Super-Resolution"

_sensors, 2023, doi:10.3390/s23052723_

Round 1

Reviewer 1 Report

In this paper, the authors propose a fully transformer-based network for depth map super-resolution. A cascaded transformer module extracts deep features from a low-resolution depth. It incorporates a novel cross-attention mechanism to seamlessly and continuously provide guidance from the colour image into the depth upsampling process. Using a window partitioning scheme, linear complexity in image resolution can be achieved,

so it can be applied to high-resolution images. The proposed method of guided depth super-resolution outperforms other state-of-the-art methods through extensive experiments. Personally, this paper is well organized and the results are complete and convincing. My comments for this paper are listed as follows.

1.     In the Introduction and Related Work, please clarify the advantages of transformer-based methods in the field of depth super-resolution.

2.     In section 3.2.3, it is mentioned that” The two significant differences are; 1. We use a cross-attention mechanism instead of self-attention, and 2. we compute the cross-attention locally for each window.”

3.     Please explain the advantages of doing this, preferably supported by experiments.

4.     In section 4.2, “We construct the CAGM module in the proposed architecture by stacking K = 6 CATBs. Each CATB is constructed from L = 6 CATL modules as described in Sec. 3.2.2.  Please explain how K and L values are determined.

5.     The experimental comparison has been verified with some state-of-the-art methods. However, the results should be analyzed in-depth and with more insightful comments on the behaviour of your algorithm. 

6.     The paper has some typos and misnomers. The authors are suggested to do the proofreading carefully.

To sum up, the above contents should be modified before publication.

Reviewer 2 Report

1. Replace ours with "Proposed" in tables and other places.

2. Fig 1, and other places, what is the name of the proposed architecture?

3. Sample images can be added in section 3.

4. Dataset section can be added with number of images, size or resolution of images etc.
